# Cleavable Bio-Based Epoxy Matrix for More Eco-Sustainable Thermoset Composite Components

**DOI:** 10.3390/polym17010088

**Published:** 2024-12-31

**Authors:** Ilaria Rossitti, Arianna Bolis, Matteo Sambucci, Fabrizio Sarasini, Jacopo Tirillò, Marco Valente

**Affiliations:** 1Department of Chemical Engineering, Materials, Environment, Sapienza University of Rome, 00184 Rome, Italy; ilaria.rossitti@uniroma1.it (I.R.); matteo.sambucci@uniroma1.it (M.S.); jacopo.tirillo@uniroma1.it (J.T.); 2INSTM Reference Laboratory for Engineering of Surface Treatments, UdR Rome, Sapienza University of Rome, 00184 Rome, Italy

**Keywords:** bio-based thermoset, ecological recycling of thermoset, ecofriendly epoxy system, composite laminates, fibers recovery

## Abstract

Cleavable bio-based epoxy resin systems are emerging, eco-friendly, and promising alternatives to the common thermoset ones, providing quite comparable thermo-mechanical properties while enabling a circular and green end-of-life scenario of the composite materials. In addition to being designed to incorporate a bio-based resin greener than the conventional fully fossil-based epoxies, these formulations involve cleaving hardeners that enable, under mild thermo-chemical conditions, the total recycling of the composite material through the recovery of the fiber and matrix as a thermoplastic. This research addressed the characterization, processability, and recyclability of a new commercial cleavable bio-resin formulation (designed by the R-Concept company) that can be used in the fabrication of fully recyclable polymer composites. The resin was first studied to investigate the influence of the different post-curing regimes (room temperature, 100 °C, and 140 °C) on its thermal stability and glass transition temperature. According to the results obtained, the non-post-cured resin displayed the highest T_g_ (i.e., 76.6 °C). The same post-curing treatments were also probed on the composite laminates (glass and carbon) produced via a lab-scale vacuum-assisted resin transfer molding system, evaluating flexural behavior, microstructure, and dynamic-mechanical characteristics. The post-curing at 100 °C would enhance the crosslinking of polymer chains, improving the mechanical strength of composites. With respect to the non-post-cured laminates, the flexural strength improved by 3% and 12% in carbon and glass-based composites, respectively. The post-curing at 140 °C was instead detrimental to the mechanical performance. Finally, on the laminates produced, a chemical recycling procedure was implemented, demonstrating the feasibility of recovering both thermoplastic-based resin and fibers.

## 1. Introduction

Today, thermosetting matrix composite materials are used in countless industrial sectors and applications, including the civil and transport industries, aeronautical, aerospace, and automotive fields, the military sector, and the production of sports equipment. These materials are recognized to be reliable solutions for high-performance and light-weight applications due to their retained dimensional stability, chemical inertia, and rigidity over a wide range of temperatures. However, once fully cured, they cannot be easily reshaped or reprocessed, thus leaving still unsolved the issues of recycling and the lack of technological flexibility [1]. These properties derive from the presence of cross-links in the molecular structure of thermosets, which have the same energy level as polymerization bonds. Both are covalent bonds, conferring on thermosetting materials high thermal and mechanical stability. However, this stability makes thermosets difficult to recycle, so current recycling technologies do not align well with modern concepts of material circularity and eco-design [2]. The primary challenge in recycling thermosetting composites lies in the degradation of the matrix and the incomplete recovery of reinforcing fibers. The leading recycling methods for thermosetting composite materials are mechanical, thermal, and chemical recycling [3,4,5]. Mechanical recycling involves several steps (crushing, grinding, milling) aimed at reducing the size of the waste composite material. These resulting fractions can be used as fillers in short fiber composites, suitable for producing sheet molding compounds. However, the equipment often suffers damage from the friction with the composite fibers, increasing the process’s operating costs. Although mechanical recycling has advantages such as non-toxicity and the ability to work at room temperature, the main issue is the generation of dust, which poses health and safety risks to operators. Thermal recycling uses high-temperature treatment or fluids to degrade the thermoset matrix and recover the fibers through processes such as pyrolysis or using a fluidized bed. Pyrolysis occurs in an inert gas atmosphere without oxygen, heating the composite to 400–1000 °C [6] to recover fibers. The matrix degrades, producing gas, oil, coal tar, and dust, necessitating fiber cleaning post-process. Fluidized bed thermal recycling, still under development, poses challenges due to polluting gases, organic solvents, and high energy requirements. Rapid heating degrades the matrix, separating it from the fibers by friction. Chemical recycling, including supercritical or subcritical solvolysis, degrades the thermoset matrix using various solvents and process variables like temperature and pressure, often with catalysts. This method yields long fibers with minimal surface residue and minor degradation in mechanical properties compared to virgin reinforcement. For instance, solvolysis can be performed at atmospheric pressure, minimizing fiber damage and retaining up to 98% of the initial fibers’ tensile strength. Although the processing temperatures involved are lower than pyrolysis, it requires expensive, corrosion-resistant equipment, and solvents can harm the environment and plant personnel. The drawback of joining these recycling routes is the lack of recovering the polymer matrix [4].

In recent years, industry and academic research teams addressed the study on the design of thermoset systems with built-in recyclability by modifying the polymer matrix with labile linkage, which makes the materials much easier to break down while retaining their mechanical characteristics [7,8,9]. Specifically, the approach is based on introducing in the polymer network degradable cross-linkers or converting permanent cross-linked structures into dynamic cross-linked ones to achieve de-crosslink and re-crosslink by exchange reactions of cleavable bonds. Indeed, the dynamic bonds implemented into the network are stimuli-responsive to specific conditions (heat, irradiation, chemical attack, or a combination of that) but not under severe conditions, resulting in the recovery of the original monomers or simpler polymers that are readily soluble [8,9]. Then, thermosets containing cleavable bonds find attractive usage for the manufacturing of fiber-reinforced polymer composites in which fibers could be easily recovered after the resin removal, preserving good thermo-mechanical characteristics for new applications [10,11,12]. Si et al. [10] proposed an epoxy vitrimer with a high concentration of exchangeable aromatic disulfide crosslinks to accelerate disulfide bond exchange reactions within the networks and enhance the recycling efficiency of epoxy vitrimers and composites. The dual disulfide vitrimer was degraded by a dilute solution of dithiothreitol (DTT). Furthermore, composites were manufactured, and the carbon fibers were reused to form new carbon fiber-reinforced polymer (CFRP) composites. However, these techniques require high-temperature conditions, which increase the cost and complexity of the recycling method. Liu et al. [11] investigated a novel epoxy network cross-linked by imine bonds and hydrogen bonds. The recycling process occurred under non-aggressive conditions at 50 °C. The fibers were recycled non-destructively and reused to prepare new composite materials, while the recoverable monomer from the matrix resin was used to cure the brittle material. Pastine [12] highlighted a series of acetal-containing amine hardeners (Recyclamine^®^, patented by Connora Technologies [7]) to cross-link commercially available epoxy resins for use in composite materials. Recyclamine^®^ hardeners have a generic polyamine structure with an innovative feature: separable terminal groups held together by a central cleavage point with acetal bonds. These bonds degrade in mild acidic solutions at certain temperatures, allowing the recovery and reuse of the remaining linear polymers. This enables the creation of cross-links that can later be broken, converting the thermoset epoxy matrix back into a reusable thermoplastic polymer (Figure 1). These novel hardening systems were implemented by other researchers to produce fully recyclable fiber-reinforced composites via resin transfer molding [13]. Saitta et al. [14] analyzed two different commercial bio-based thermoset resins to design flax fiber-reinforced composites showing recycling characteristics. The authors studied the optimum process parameters to be implemented for the composite preparation as well as the recyclability yield through a specific chemical protocol, verifying the feasibility of separating and recovering both the fibers and resin from the composite samples.

Research efforts on the design, synthesis, and processing of fully recyclable thermosetting formulations are continually expanding, with the major target of implementing these polymeric systems in industrial sectors of high technological value. For instance, these recyclable thermosets are considered promising directions of development in the wind energy industry for producing novel recyclable wind turbine blades [15], providing comparable mechanical performance with respect to traditional epoxy resin composites and the recyclability characteristic of thermoplastics [16]. To the best of the authors’ knowledge, new recyclable epoxy systems are emerging to address the upcoming market demand and in-depth investigation is needed regarding processability for composite material fabrication as well as recyclability aimed at recovering fibers and matrix to reintroduce in new applications.

The present work addressed the recycling pathway of a new cleavable bio-based epoxy resin system (R-Concept, Barcelona, Spain) implemented in the fabrication of fully recyclable composite materials. These bio-epoxy systems combine double benefit towards more sustainable products: (1) incorporation of a bio-based component in the resin formulation to reduce the environmental impact associated with the use of toxic and expensive fossil sources as feedstocks and (2) implementation of a Recyclamine^®^-type hardener that enables the recovery of reinforcement (carbon fibers, glass fibers) and the epoxy matrix as recyclable thermoplastic, both further integrable into new processes and applications. The resin formulation was first characterized by thermal analysis to evaluate the effect of different curing conditions on the matrix performance. Carbon and glass fiber-reinforced composites were then fabricated via vacuum-assisted resin transfer molding. Thermal, mechanical, and microstructural characterization was performed on the composite materials obtained. Finally, an experimental chemical recycling protocol was developed for the recovery of fibers and the thermoplastic fraction.

## 2. Materials and Methods

### 2.1. Materials

The Beluga Whale epoxy resin and the Recyclamine-type hardener R*LAB 005, both provided by R-Concept (Barcelona, Spain), were used as the thermoset matrix for the composites. The manufacturer specifies that R*LAB 005 is an “extra slow” hardener, with a gel time of 100–130 min at 60 °C. While detailed chemical structures for these materials are not yet available in the literature, they are part of the bio-based epoxy resin and Recyclamine™ hardener family, known for enabling the recyclability of thermoset systems.

Previous studies [17] have characterized the Recyclamine™ R*101 hardener, which contains cleavable ketal groups within the cross-linked network. These groups can be selectively broken under mildly acidic conditions, allowing for the depolymerization of the cured matrix and the recovery of both the polymer and reinforcing fibers. The chemical structure of R*101, shown in Figure 2a, serves as a representative example of this class of materials. Similarly, Recyclamine™ R*LAB005 can be assumed to share this functional mechanism, as it belongs to the same family of cleavable hardeners. Furthermore, the Beluga Whale resin shares similarities with the Polar Bear resin described in the literature [4], which contains 28% bio-based carbon (expressed as a fraction of total organic carbon), as shown in Figure 2b. By referencing these well-characterized systems, such as Recyclamine™ R*101 and Polar Bear, we provide a solid framework to understand the chemical behavior and recyclability of the Beluga Whale/Recyclamine™ R*LAB005 system studied here.

Carbon and glass fabrics were used as reinforcement for different composites. The carbon fabric was a plain weave with an areal density of 160 g/m^2^, while the glass fabric was a twill weave with an areal density of 290 g/m^2^. Both fabrics were supplied by Angeloni Group S.r.l. (Venice, Italy).

### 2.2. Resin System Formulation

The resin-to-hardener ratio was set to 100:35 as recommended by the R-Concept resin manufacturer. Furthermore, this bio-epoxy system cures at room temperature, and post-cure options are not indicated in the technical datasheet. Therefore, given the lack of studies in the literature on this resin and to establish the best processing parameters for the subsequent manufacturing of composites, three post-curing conditions were investigated: no post-curing (RT), 100 °C for 3 h (PC100), and 140 °C for 3 h (PC140). Similar post-curing treatments were probed by Ferrari et al. [18], who investigated R-Concept resins in the production of biocomposites incorporating waste flour. The resin was poured into a steel mold to manufacture specimens with dimensions suitable for the thermal analysis.

### 2.3. Composite Laminates Manufacturing

Composite laminates (200 mm × 150 mm) were realized using a lab-scale vacuum-assisted resin transfer molding system; see Figure 3a. System schematization and auxiliary materials used in the infusion process are shown in Figure 3b. The infusion layout consisted of: (a) a glass mold surface, (b) the fiber reinforcement stack (10 layers with 0/90 orientation), (c) a polyethylene perforated release film to control the resin content into the laminate (Flow mesh), (d) a polyester fabric bleeder cloth to allow airflow throughout the vacuum bagging process as well as bleed out an excess resin in a composite part (Breather), and (e) a polytetrafluoroethylene (PTFE) film sealed to the mold by a tackifier tape to form a sealed chamber (Vacuum bag). Silicon tubes were used to let the resin enter the bag and to allow the suction of the air, and then the resin itself. The system was connected to a Divac 0.6L diaphragm pump (Leybold, Cologne, Germany) to ensure a relative vacuum pressure of −0.8 bar. For each type of fabric, three composite plates have been produced, one for each post-curing condition implemented for the neat resin (RT, PC100, and PC140). The obtained glass and carbon composites had an average thickness of 2.5 mm. Specimens for thermal and mechanical characterization were obtained from the plates by means of a computerized numerical control (CNC) cutting machine (Falcon 1500 by Valmec, Pescara, Italy). The carbon and glass fiber volume fractions were determined using a commercial density measurement kit paired with a 0.1 mg resolution analytical balance based on the principle of hydrostatic thrust, Mettler Toledo ME54 (Mettler Toledo, Worthington, OH, USA). The glass and carbon fiber-reinforced composites exhibited fiber volume contents of 47% and 36%, respectively.

### 2.4. Procedure for Composites Recycling: Fiber and Polymer Recovery

After the thermal and mechanical characterization of the composite laminates, the materials were subjected to a recycling process following the experimental protocol implemented by Cicala et al. [19]. The first step is an acid attack, where about 10–11 g of composite material is immersed in 300 mL of acetic solution (40 vol.% of acetic acid). Acetic acid (purity level ≥ 99.8%) was purchased from Honeywell. The solution is brought up to 80 °C and left to react for 2.5 h. Once the thermoset matrix is dissolved, the fibers are filtered from the solution, then washed in distilled water and left to dry. At this point, the acid solution, remaining at a temperature of 80 °C, is neutralized with 105 mL of a 2 M aqueous solution of NaOH (Carlo Erba Reagents S.A.S., Val-de-Reuil, France). By introducing the basic solution into the acid, therefore, a precipitate forms progressively. That corresponds to the thermoset matrix recovered as a thermoplastic polymer. Once all the basic solutions have been introduced with a pipette, the system is left to react and cool to a temperature of 40 °C. Next, the thermoplastic is filtered from the solution and washed with 150 mL of distilled water. A few drops of basic solution are added to aggregate the precipitates, always using a pipette. In the end, the polymer obtained is filtered and left to dry. The workflow of the recycling procedure is illustrated in Figure 4.

### 2.5. Testing Methods

#### 2.5.1. Thermogravimetric Analysis (TGA)

TGA was performed on both the bio-epoxy neat matrix and the recovered polymer from the recycling process using a TG 209 F1 Libra analyzer by Netzsch (Selb, Germany). The samples (~20 mg) were tested in an inert nitrogen atmosphere from room temperature to 800 °C, employing a heating ramp of 10 °C/min.

#### 2.5.2. Differential Scanning Calorimetry (DSC)

DSC was carried out with a DSC 214 Polyma by Netzsch (Selb, Germany) in a nitrogen atmosphere, with a temperature range from −40 °C to 170 °C and a scanning rate of 20 °C/min. The mass of the tested samples (bio-epoxy neat matrix at the different post-curing conditions and the recovered polymer) was ~10.5 mg. The glass transition (T_g_) values were recorded on the second heating scan to avoid interference from the endothermic peak of the polymer’s relaxation enthalpy. The relaxation enthalpy occurs in the glass transition region and corresponds to the release upon heating of degrees of freedom after the material has been cooled below its T_g_ [20].

#### 2.5.3. Mechanical Testing on Composite Laminates

Hardness and flexural tests were conducted to evaluate the influence of the type of fabric (glass and carbon) and the post-curing regime on the mechanical behavior of the manufactured laminates.

Shore D hardness measurements were performed using an analog hardness tester (Zwick/Roell GmbH, Ulm, Germany) according to ASTM D-2240 [21]. The hardness value for each sample was calculated as an average of ten measurements.

Three-point flexural tests were conducted on a Zwick/Roell Z010 machine (Zwick/Roell GmbH, Ulm, Germany), following the standard method ASTM D7264 [22]. Samples (100 mm length and 10 mm width) were tested at 2 mm/min with a span of 70 mm. The strain was recorded with a displacement transducer in contact with the samples. At least three replicates were performed on each formulation.

#### 2.5.4. Dynamic Mechanical Analysis (DMA)

Following the study of the mechanical properties of the composite laminates, DMA was conducted considering the optimum post-curing condition. The analysis was performed according to ISO 6721 [23] with a DMA 242 E Artemis by Netzsch (Selb, Germany) in a three-point bending configuration from 30 °C to 140 °C using a heating ramp of 2 °C/min and a frequency of 1 Hz.

#### 2.5.5. Scanning Electron Microscopy (SEM)

The fiber-matrix interface in the produced composite laminates as well as the surface morphology of the fibers recovered from the recycling process were analyzed by means of SEM (Tescan Mira 3, Brno, Czech Republic). Prior to the investigation, the specimens (cross-section of the composite specimens and recovered fibers) were gold-coated using an Edwards S150B sputter-coater (Edwards Ltd., Burgess Hill, UK).

## 3. Results

### 3.1. TGA on Bio-Epoxy Resin Matrices

The thermogravimetry (TG) and derivative thermogravimetry (DTG) curves of the resin post-cured under different conditions (RT, PC100, and PC140) are reported in Figure 5. The graph clearly shows that the weight loss occurs in two steps. The first degradation stage was concentrated in the 180–200 °C temperature range with a mass change of about −5 wt.%. Within the temperature range of 340–370 °C, the main degradation event was observed, with a massive loss in weight of about 80%. In this stage, the thermal decomposition took place for the bio-epoxy resins. Table 1 summarizes the maximum temperature of the two degradative processes undergone by the resin. T_I_ stands for the maximum temperature related to the first weight loss, whereas T_max_ refers to the maximum temperature at which the main degradation process occurs. TGA results highlighted that the non-post-cured resin showed slightly superior thermal stability compared to the cured ones. Furthermore, as the post-cure temperature increases, the thermal stability of the polymer decreases. This result can be explained by considering the thermo-degradative effects that the resin experiences due to the presence of a “bio” component in its chemical structure. Indeed, these types of polymers have a substantial portion of the carbon content replaced by a biomass origin that is sensitive to high-temperature stress [24]. In agreement with the DTG analyses reported by di Mauro et al. [25] and considering the chemical structure of the R-Concept bio-epoxy system [4], bio-based thermosets follow a two-step thermal degradation mechanism, as also reported by Supanchaiyamat et al. [26]. The first degradation step is attributed to the breakage of ester linkages in the polymer’s crosslinked structure. These linkages, introduced during the curing process through the reaction between epoxy groups and functional groups of the hardeners, are chemically labile and break at low temperatures. This process leads to the formation of low molecular weight intermediate fragments. The second degradation step, on the other hand, is related to the thermal cleavage of these intermediate fragments, occurring at higher temperatures. This phenomenon represents the complete disintegration of the polymer network, causing a rapid increase in mass loss. The presence of two distinct degradation events is clearly visible in the DTG curves [25] and allows for the identification of the contribution of each phase to the overall thermal behavior of bio-based thermosets.

This behavior, although like that of systems containing disulfide dynamic bonds, highlights lower thermal stability compared to resins crosslinked with more rigid dicarboxylic acids or longer carbon chains [27,28]. However, the statistical heat-resistant index temperature values (Ts) [25] of bio-based thermosets with low epoxy content are higher than those of resins such as epoxidized sucrose soyate crosslinked with citric or malic acids [29]. This apparent discrepancy reflects the ability of some bio-based systems to better withstand initial temperatures before degradation begins, despite showing lower overall thermal stability compared to systems crosslinked with more rigid hardeners.

### 3.2. DSC on Bio-Epoxy Resin Matrices

Figure 6 shows the DSC curves, with (a) representing the first heating cycle and (b) the second one. T_g_ values from both cycles are listed in Table 2: the first cycle provides insight into the initial state of the resins, while the second cycle values, used for the analysis, represent their stabilized thermal properties. Comparing the first and second cycles, we observe that additional curing reactions occur during the first heating for the non-post-cured (RT) and PC100 samples. For the RT sample, the T_g_ increases from 69.7 °C in the first cycle to 76.6 °C in the second cycle, indicating that the resin undergoes further curing during the first heating. A similar trend is observed for the PC100 sample, with the T_g_ increasing from 68.0 °C in the first cycle to 72.7 °C in the second cycle. In contrast, for the PC140 sample, the T_g_ remains almost unchanged (51.7 °C in the first cycle versus 50.9 °C in the second cycle), suggesting that the polymer network is already significantly degraded due to the high post-curing temperature. These results confirm that the first heating cycle is critical for understanding the thermal behavior and cure state of the resins. The significant increase in T_g_ between the first and second cycles for RT and PC100 samples indicates that their polymer networks are not fully stabilized initially. Referring to the second table, the non-post-cured resin had the highest T_g_ value (76.6 °C). By implementing thermal post-cure treatments, the T_g_ was progressively reduced. The PC100 condition did not significantly alter the thermal characteristics of the material (approximately 4 °C lower than the RT condition), while at 140 °C post-curing a drop of almost 25 °C compared to the RT sample was found. The results confirm the evidence highlighted by the TGA analysis, namely that thermal stress due to excessive post-curing temperature induces degradation of the biomass-deriving fraction of the resin, resulting in T_g_ decline. The influence of post-curing conditions on the thermo-mechanical properties of bio-based epoxy resins was investigated by Lascano et al. [30]. The authors found that excessively high temperatures potentially restricted the formation of homogeneously cross-linked structures by limiting the molecular diffusion during the resin curing, resulting in a wide distribution of gaps between the cross-linked regions. As a result, a reduction in T_g_ was detected due to the increased free volume fraction in the polymeric network. The T_g_ values obtained under RT and PC100 conditions are consistent with the range declared by the supplier (55–85 °C). This bio-resin system that achieves optimal thermal performance without special thermal curing conditions can certainly be valuable for industrial mass production reasons and for fabricating large-sized components (e.g., wind turbine blades).

### 3.3. Mechanical Testing on Composite Laminates

Figure 7 displays the effect of different post-curing regimes on the flexural behavior of carbon and glass composite laminates. As expected, the carbon laminates outperformed the glass-based ones. From Figure 7a, the carbon fiber-reinforced laminate showed higher flexural strength for RT and PC100 conditions (607 MPa and 624 MPa, respectively). The flexural strengths of glass fiber-reinforced composites were 510 MPa and 569 MPa for RT and PC100 post-curing, respectively. Therefore, compared to the neat matrix, the post-cure treatment at 100 °C on composites was effective in improving the mechanical properties. Although the insertion of the fiber layer reduces the macromolecular mobility and the proper matrix crosslinking degree, a well-balanced post-cure treatment would assist the increment in crosslink density and interfacial bonding with the fibers [31]. However, the post-cure treatment at 140 °C was excessive, leading to a reduction in flexural strength (–10% for glass laminate and −23% for carbon laminate over to the composites post-cured under PC100 condition). Too high post-curing temperatures would cause excessive resin shrinkage and the development of residual stresses at the fiber-matrix interface (debonding) [32]. PC140 treatment degraded carbon-based laminates more significantly than glass-reinforced ones. It is plausible that this effect is also associated with the different degrees of compaction experienced by the two types of fabrics under the same processing conditions used to produce the composites. The SEM investigation in the next section will clarify this aspect. As shown in Figure 7b and on the stress-strain plots displayed in Figure 7c,d, post-cure has a negligible effect on the laminate stiffness in their corresponding groups. This evidence agrees with the one reported in earlier works by Umarfarooq et al. [31].

Shore D hardness of the laminates (Figure 8) was weakly influenced by both the type of reinforcement and the post-cure treatment. However, the results show a slight improvement in hardness in the case of PC100 laminates, corroborating the flexural test results. An optimal post-curing regime improves the bonding between fibers and resin, due to an increase in cross-linking and stacking, which, although generally reducing polymer chain mobility, may also lead to an increment in T_g_, thus enhancing resistance to the penetration of the indenter [33].

### 3.4. SEM Analysis on Composite Laminates

SEM micrographs in Figure 9 illustrate the cross-section surface of the glass (Figure 9a,c) and carbon (Figure 9b,d) laminates post-cured at the best condition (PC100).

In Figure 9c, the scale bar differs from the others due to the higher magnification used for this image, providing a more detailed view of the specific features of the material. At the same post-curing regime, the laminates showed marked differences in terms of microstructure. The glass-based composite displayed a non-defective cross-section free of inter-layer and inter-fiber voids. The carbon-based composite laminate, although over the PC100 post-cure conditions had superior mechanical performance compared to glass-reinforced composite, showed several defects dispersed throughout the surface, which limited the potential improvement of carbon fibers on the strength of the final composite. Figure 9d highlights more evident delamination of the composite due to the poor compaction of the fabric. The formation of voids in the composite is attributable to two main aspects, already discussed in the review work by Xueshu and Fei [34]: vacuum pressure and resin flow. Under the pressure conditions implemented in the present study, the carbon fabric, being significantly more rigid than glass, provided greater resistance to the compaction and resin flow, resulting in non-impregnated zones inside the laminates. Therefore, optimization of the process parameters of resin transfer molding (the pressure) is challenging to increase the performance of carbon-based composites by minimizing the percentage of voids.

As verified by mechanical tests, post-cure treatment at 140 °C significantly degraded the performance of the carbon-based laminate. The SEM micrograph of Figure 10 shows the fracture surface of the PC140 carbon-reinforced composite specimen, highlighting extensive fiber pull-out and interfacial debonding, indicative of poor fiber-matrix adhesion caused by the excessive post-curing temperature. This deterioration in interfacial bonding compromises the load transfer efficiency between the matrix and the fibers, thereby reducing the overall mechanical properties of the composite. Fiber pull-out occurs when composites are post-cured at elevated temperatures above the glass transition of the matrix. As the polymer becomes critically softer, adhesive and cohesive failures turn out to be more dominant. This phenomenon then allows for fiber and matrix materials to be more easily debonded [35].

### 3.5. DMA on Composite Laminates

Thermo-mechanical properties of the composites were studied by DMA on the samples post-cured at 100 °C (PC100), representing the laminates with the highest mechanical strength properties. Plain resin post-cured at the same condition was taken as a reference to evaluate the change in T_g_ as a function of the type of reinforcing fabric and fiber-matrix interaction. Variations of the damping factor (tan δ) with respect to temperature curves are compared in Figure 11. T_g_ was then evaluated as the peak temperature of the curve, and the inherent values were summarized in Table 3. As might be expected, the composites lead to a shift of T_g_ towards a higher temperature compared to pure resin because of the lower molecular mobility of the polymer chain due to the presence of fibers [36]. However, the extent of the increase in the glass transition is quite different between carbon and glass-based laminates, reflecting different fiber-matrix interfacial properties experienced in the two materials. The microstructural defectiveness found in carbon-based laminates was responsible for a limited increase in T_g_ (about 1 °C higher than pure resin). On the other hand, glass composite showed a more marked increment (+4 °C) justifying the homogeneous fiber impregnation and better interfacial fiber-matrix adhesion. The different fiber-matrix bonding was also interpreted in terms of tan δ values. The damping factor is a sensitive indicator of all kinds of molecular motions that are going on in a material. Logically, in a composite, the molecular motions at the interface contribute to the damping of the material. The strong cohesion of fibers and polymer matrix reduces the mobility of the molecular chains at the interface and therefore brings a damping reduction [37]. The lowest tan δ peak value detected in the glass-based composite sample corroborated the evidence of better interfacial adhesion between glass fiber and resin.

### 3.6. Chemical Recycling on Composite Laminates

The chemical recycling procedure implemented in the present work showed material recovery yields exceeding 95%. As detailed in Table 4, it was feasible not only to recover clean fibers suitable to produce new fabrics from recycled feedstock (such as woven mats) for composites but also possible to convert the thermoset matrix into a reprocessable thermoplastic polymer. Furthermore, the recycling process has an extremely low environmental impact, both in terms of energy consumption and in terms of pollutants. In fact, the maximum temperature achieved is about 80 °C, and the chemical additives employed are acetic acid and sodium hydroxide. In addition, acetic acid could be recovered and used for further chemical recycling stage or used in other industrial processes involving sectors such as food, pharma, chemical, textile, polymer, medicinal, and cosmetics [38]. Such an aspect would lead to a totally eco-sustainable production and disposal of thermosetting matrix composite materials.

#### 3.6.1. Thermal Characterization of the Recovered Polymer

Figure 12 reports the TGA thermogram of recovered polymer from composite laminate post-cured at 100 °C. Thermal degradation of the polymer occurred in two regions. The initial mass loss (10%) was related to the release of moisture remaining trapped in the polymer’s structure during drying after recycling treatment. The main degradation took place around 340 °C (onset temperature of 315.3 °C) due to the total decomposition of the recovered thermoplastic. The result agrees with those from Dattilo et al. [17], who studied the thermomechanical properties of thermoplastic polymers deriving from an R-Concept bio-based epoxy resin system (Polar Bear + R*101) recovered from a similar chemical recycling protocol. The authors ascribed the degradation to the chain scission of the C–O bond.

The observed difference in degradation profiles, where a single degradation peak is seen in the TGA analysis of the recycled thermoplastic while two distinct peaks appear for the cured epoxy resin, can be primarily attributed to the molecular structure [17]. The cured bio-based epoxy resin consists of a crosslinked network that undergoes two distinct degradation phases due to the complexity of the crosslinks [25,39]. Upon chemical recycling, this crosslinked structure is cleaved, converting the material into a simpler, linear polymer thermoplastic, which undergoes thermal degradation in a single, unified step. In Figure 13 [4,17], the chemical structure studied by Dattilo et al. [17] of the recycled polymer is shown, which was characterized using advanced techniques such as MALDI (Matrix-Assisted Laser Desorption/Ionization) spectroscopy, Gel Permeation Chromatography (GPC), and Nuclear Magnetic Resonance (NMR) spectroscopy. Dattilo et al. [17] state that two distinct chemical structures were identified for the recycled polymer, resulting from the selective cleavage of bonds in the cured network during the chemical recycling process. The recycled polymers obtained resembled poly(hydroxyaminoethers) with thermal stability up to 400 °C. Furthermore, a clear correlation was observed between the curing degree of the epoxy resin and the molecular mass of the recycled polymer, indicating that higher curing degrees lead to a recycled polymer with a greater molecular mass.

It Is worth mentioning that for the TG curve, a 20% remaining mass appears at 800 °C. This value determines that some fibers pass through the solution’s filtration system during the second step of the recycling process, remaining embedded inside the polymer mass. It is important to underline that at an industrial scale, with an optimized filtration system, the remaining small portion of fibers within the polymer can be avoided by obtaining a “clean” matrix for new processing. However, the “contamination” of the recovered thermoplastic with the fibers can also represent an added value when the aim is to obtain fiber-filled pellets intended for injection molding processing.

The DSC thermogram of the recovered polymer is displayed in Figure 14. As also verified from TGA, the first heating curve showed a broad endotherm around 100–150 °C, representing the loss of moisture from the sample. The T_g_ was determined from the second heating curve, obtaining a value of 59.4 °C. This value falls within the representative range of glass transition temperature of these polymer systems listed in Ref. [40]. As expected, the achieved T_g_ is lower than those of bio-epoxy resins due to the conversion of the network into a linear structure typical of thermoplastics. The referenced study also reported promising mechanical properties for these recovered polymers (tensile modulus of 2.4 GPa and tensile strength of 57 MPa), demonstrating good potential for new applications and competitiveness with other well-established thermoplastic polymers on the market. This will require a more extensive characterization of the polymer recovered in the present work and a related study on its processing feasibility (such as by injection molding or 3D printing).

#### 3.6.2. Morphological Characterization of the Recovered Fibers

The morphology of the fibers recovered from the chemical process was observed via SEM. Figure 15a,b show the micrographs of glass and carbon fibers, respectively. The surfaces were quite clean, with just a few polymeric residues along carbon fibers. However, by means of ultrasonic bath cleaning, most of the loosely attached residue on the surface can be easily removed. In addition, the implemented chemical treatment does not affect the fiber surface quality in terms of roughness and defects. Single-fiber mechanical characterization will be necessary to investigate in detail the influence of the recycling process on the structural integrity of the reclaimed fibers. This is a fundamental aspect to know also to direct the use of the recovered fibers in new engineering applications.

## 4. Conclusions

This paper studied the properties, processability for composite laminate manufacturing, and recyclability of a room-temperature curable and fully recyclable bio-epoxy resin system. First, the bio-based resin was thermally characterized to assess the influence of different post-curing conditions on its thermal stability and glass transition temperature. Then, glass and carbon composite laminates were produced, investigating the mechanical, microstructural, and thermal properties as a function of the selected curing regimes. Finally, a recycling procedure was implemented to verify the possibility of recovering the thermoplastic-based matrix and fibers reusable for new applications.

The main results are listed below:For the neat bio-epoxy resin, the post-cure at 140 °C progressively reduced the thermal stability and the glass transition temperature due to degradation phenomena of the bio component of the resin. The best condition was the non-post-cured resin showing a glass transition temperature of 76.6 °C.The post-curing at 100 °C on the produced composite laminates positively influenced the enhancement of the cross-linking of polymer chains, which was reflected in the improvement in the mechanical strength. With respect to the non-post-cured laminates, the flexural strength improved by 3% and 12% in carbon and glass-based composites, respectively. The post-curing at 140 °C was instead detrimental to the mechanical performance.Carbon laminates were more affected by structural defects due to the combined effect of the high stiffness of the fabric and the unoptimized pressure conditions of the resin transfer molding system. This was evident from both the SEM microstructural analysis and the glass transition analysis using DMA. The results show that, although carbon fiber composites exhibit superior mechanical performance, their glass transition temperature (T_g_) increased by only 1 °C compared to pure resin, due to microstructural defects and poor fiber impregnation. In contrast, glass fiber-based composites show a greater T_g_ increase (+4 °C), owing to better fiber-matrix adhesion and more homogeneous fiber impregnation. This highlights the importance of impregnation quality and fiber-matrix interface cohesion in determining the thermal properties of the composites.A successful chemical recycling procedure was developed accounting for recovery yields up to 98.8%. The recovered thermoplastic showed relevant thermal properties for new engineering applications (thermal stability of up to 400 °C and glass transition temperature of 59.4 °C). The fibers were recovered cleaner and without evident damage. This approach enables a circular economy scenario within the composite materials sectors.

These outcomes are encouraging for further research efforts. The next works will focus on the optimization of the composites’ manufacturing procedure to achieve the best performance and enable the possibility of creating fully recyclable components intended for a lot of industrial sectors (automotive, energy, etc.). Special focus will be on the characterization and processability of the secondary raw materials recovered from the recycling process. Reclaimed polymers and fibers can represent new “circular” feedstock within the composite industry.

## Figures and Tables

**Figure 1 polymers-17-00088-f001:**
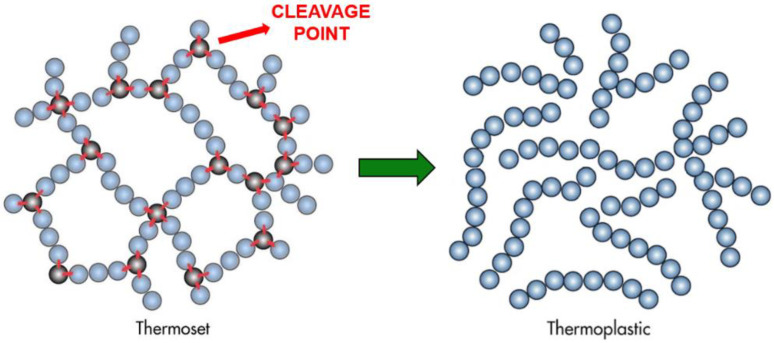
Schematic of the cleavage mechanism induced by Recyclamine^®^ hardener and conversion to the thermoplastic system (authors’ own figure).

**Figure 2 polymers-17-00088-f002:**
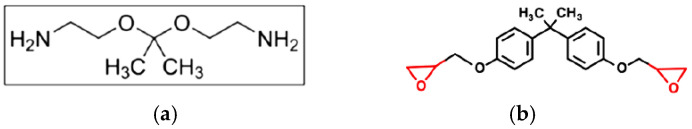
Chemical structure of (**a**) Recyclamine^TM^ R*101 [17] and (**b**) Polar Bear (part A) [4].

**Figure 3 polymers-17-00088-f003:**
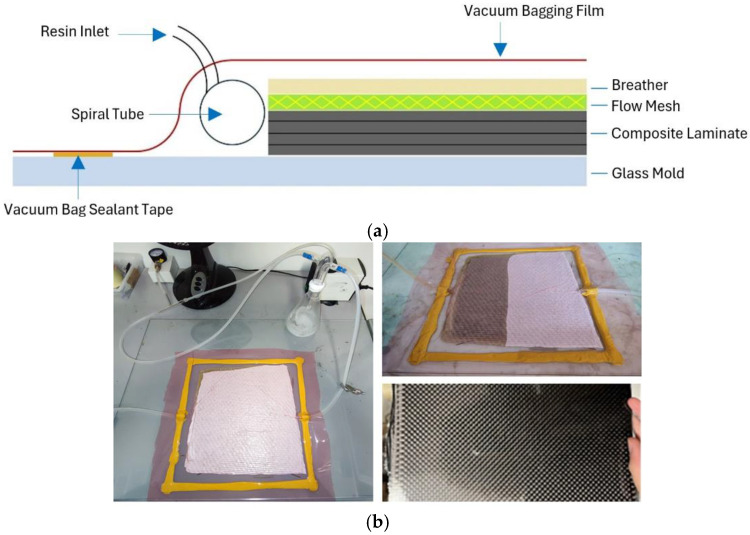
Vacuum-assisted resin transfer molding system designed for the laminates fabrication: (**a**) schematic of the system and (**b**) experimental set-up.

**Figure 4 polymers-17-00088-f004:**
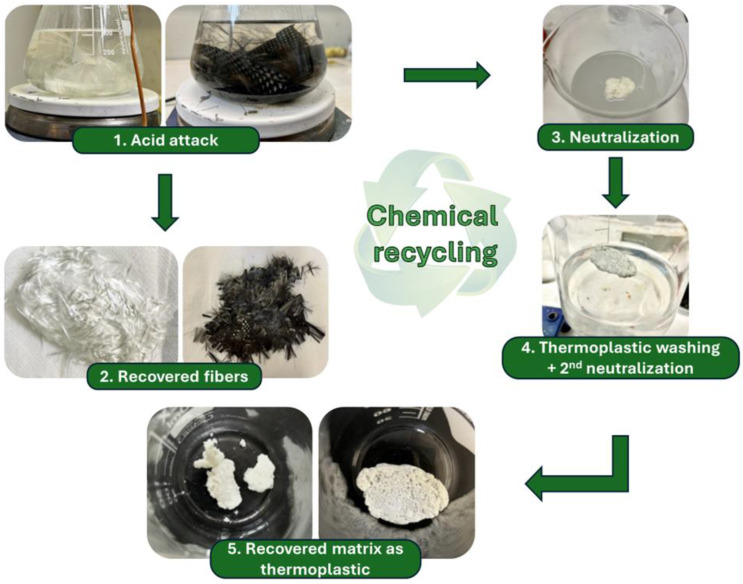
Schematization of the chemical recycling process.

**Figure 5 polymers-17-00088-f005:**
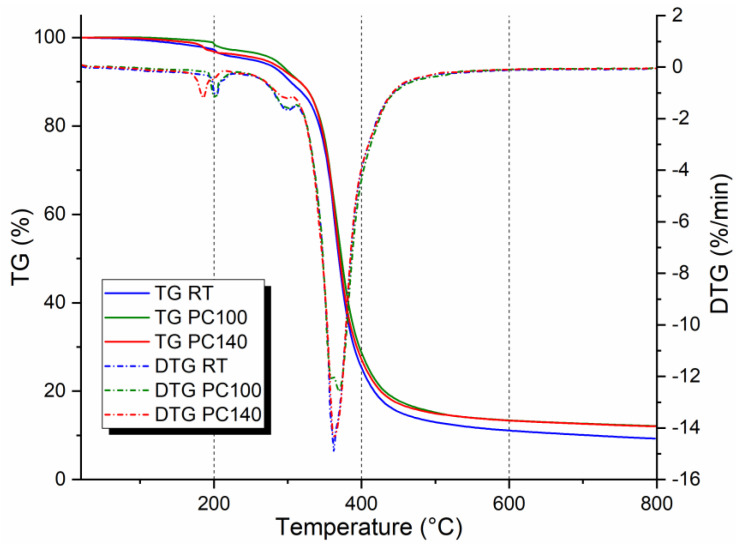
TGA curves of bio-epoxy resins post-cured at different conditions: RT, PC100, PC140.

**Figure 6 polymers-17-00088-f006:**
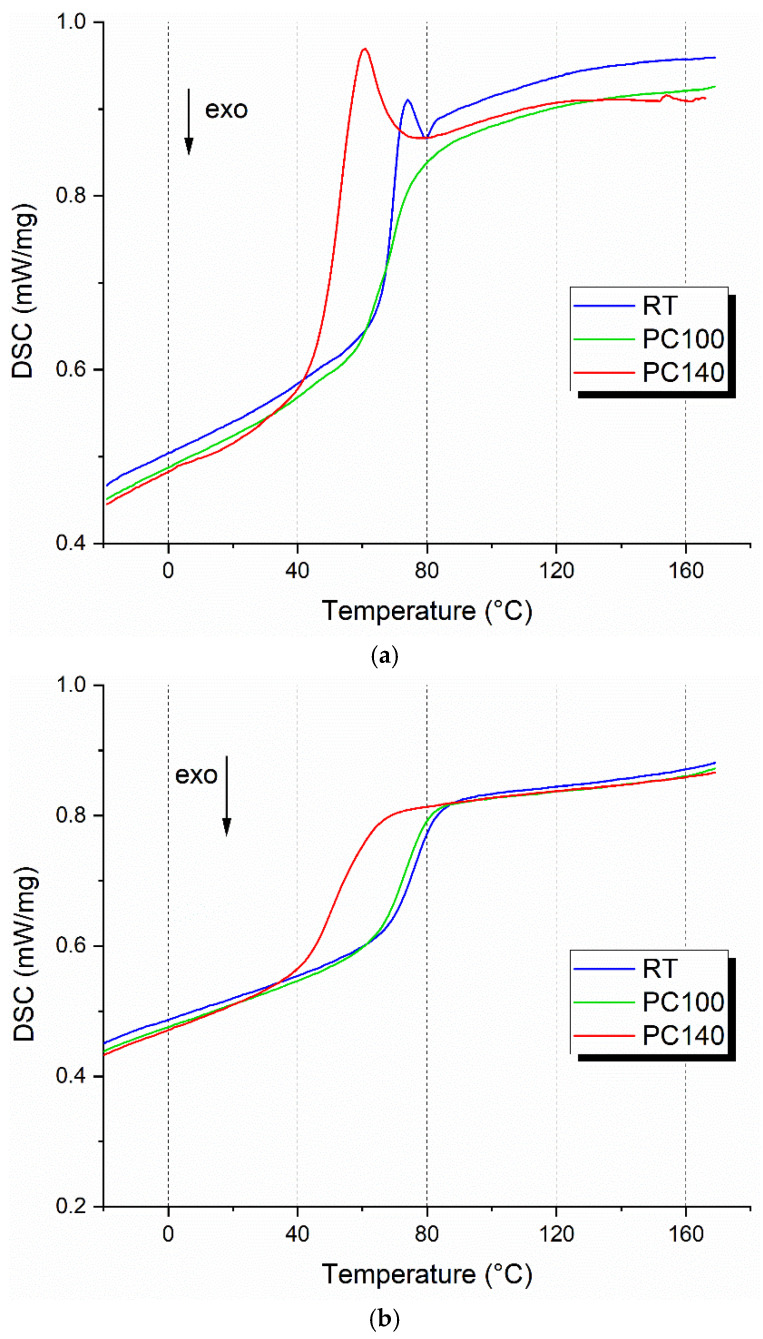
First (**a**) and second (**b**) heating scan DSC curves of bio-epoxy resins post-cured at different conditions: RT, PC100, PC140.

**Figure 7 polymers-17-00088-f007:**
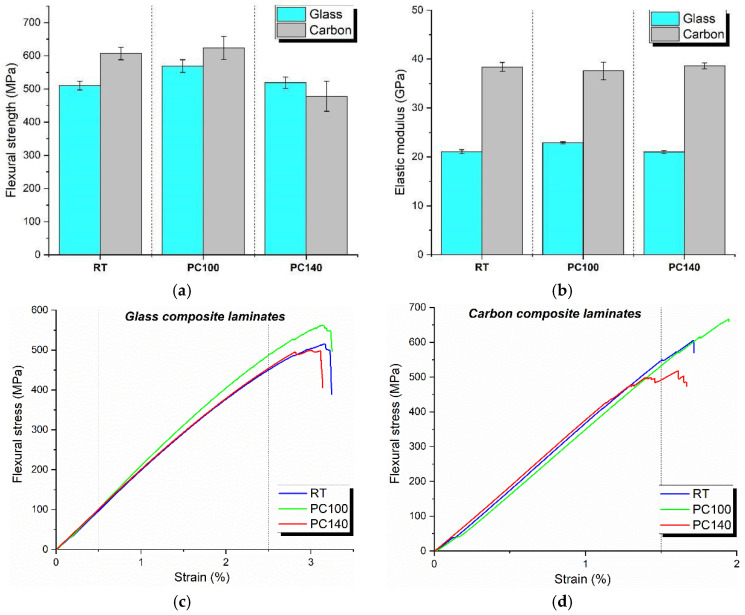
Flexural test results on composite laminates at different post-curing regimes: (**a**) flexural strength, (**b**) elastic modulus, (**c**) flexural stress-strain curve for glass-reinforced laminates, and (**d**) flexural stress-strain curve for carbon-reinforced laminates.

**Figure 8 polymers-17-00088-f008:**
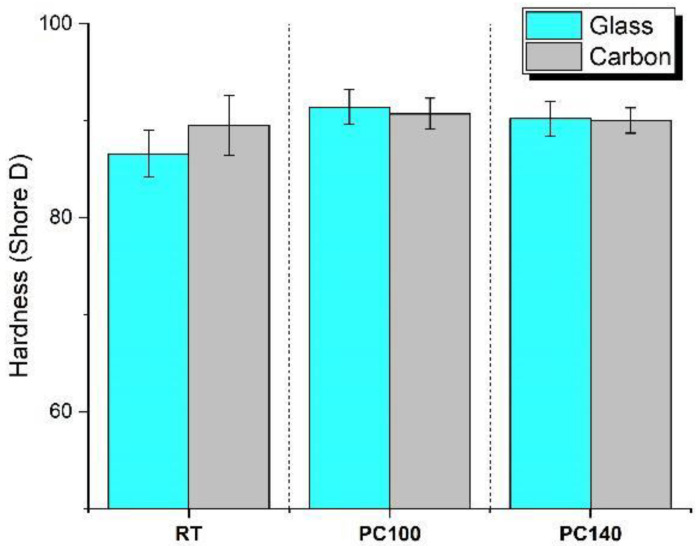
Shore D hardness test results on composite laminates at different post-curing regimes.

**Figure 9 polymers-17-00088-f009:**
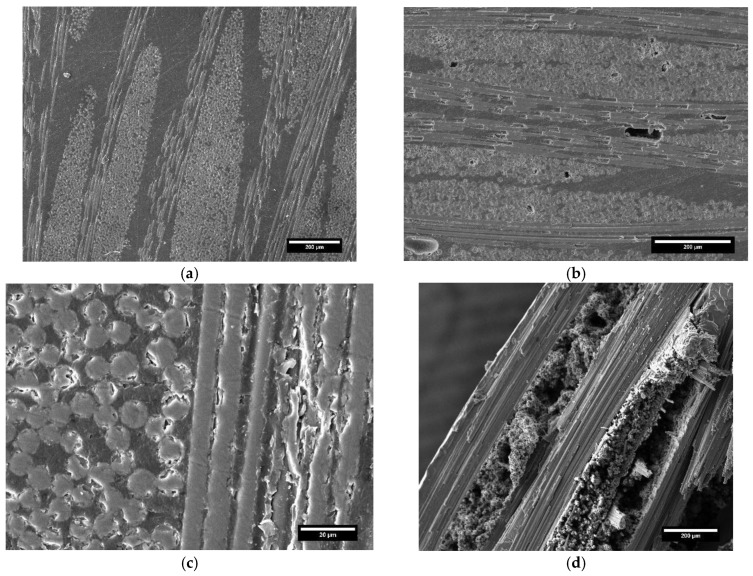
SEM analysis on the composite over PC100 condition: (**a**) glass laminate (polished surface), (**b**) carbon laminate (polished surface), (**c**) detail on the glass-resin interface (polished surface), and (**d**) fracture surface of carbon-based laminate.

**Figure 10 polymers-17-00088-f010:**
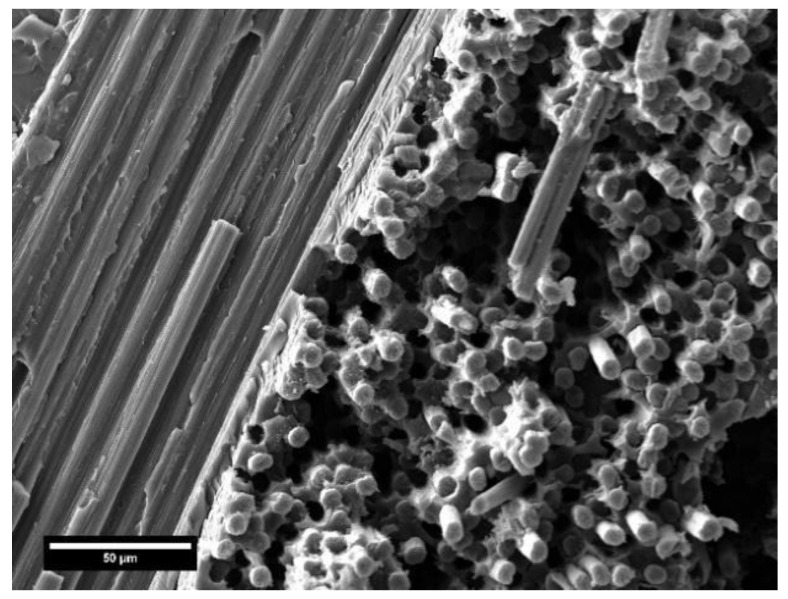
SEM analysis on the carbon-based composite laminate over PC140 condition: detail on fiber-pull-out.

**Figure 11 polymers-17-00088-f011:**
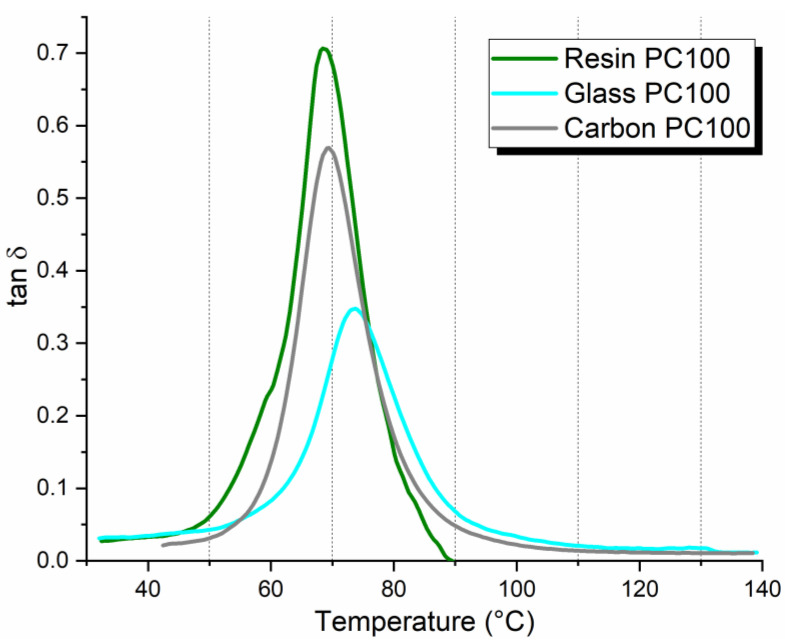
DMA test results: tan δ vs. temperature for plain resin, glass, and carbon laminates post-cured under the PC100 regime.

**Figure 12 polymers-17-00088-f012:**
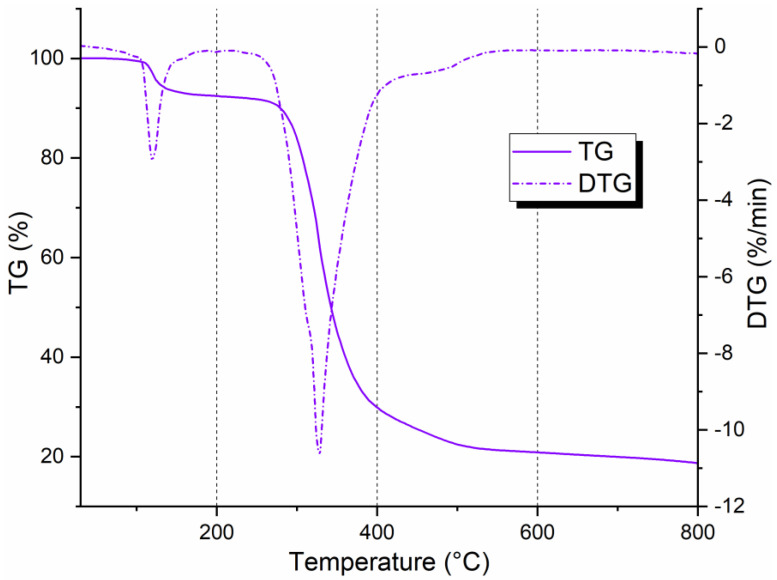
TGA on recovered polymer from chemical recycling.

**Figure 13 polymers-17-00088-f013:**
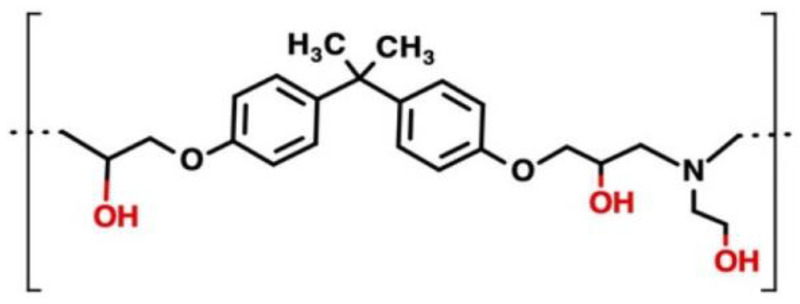
Thermoplastic deriving from the chemical recycling process of the epoxy resin matrix (Polar Bear and Recyclamine^TM^ R*101) [4,17].

**Figure 14 polymers-17-00088-f014:**
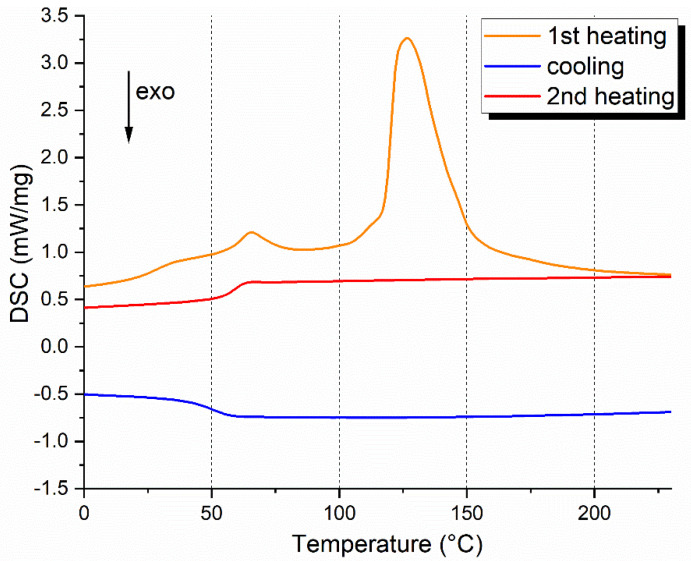
DSC on recovered polymer from chemical recycling.

**Figure 15 polymers-17-00088-f015:**
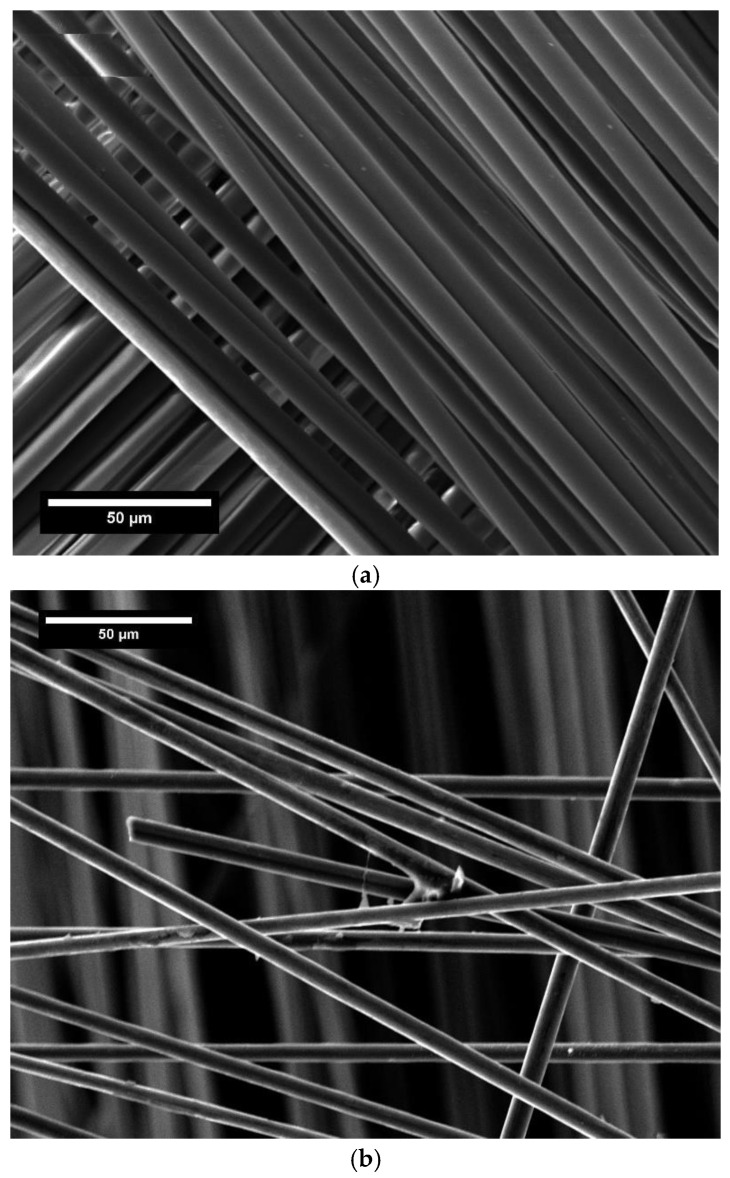
SEM analysis on recycled fibers: (**a**) glass and (**b**) carbon fibers.

**Table 1 polymers-17-00088-t001:** T_I_ and T_max_ values determined by TGA.

Post-Curing Condition	T_I_ (°C)	T_max_ (°C)
RT	203.8	366.9
PC100	202.0	367.7
PC140	185.8	362.9

**Table 2 polymers-17-00088-t002:** T_g_ values determined by DSC analysis.

Post-Curing Condition	T^1°^_g_ (°C)	T^2°^_g_ (°C)
RT	69.7	76.6
PC100	68	72.7
PC140	51.7	50.9

**Table 3 polymers-17-00088-t003:** DMA test results: T_g_ and tan δ peak-values.

Sample	T_g_ (°C)	tan δ Peak
Resin PC100	68.4	0.71
Glass PC100	73.1	0.35
Carbon PC100	69.5	0.57

**Table 4 polymers-17-00088-t004:** Results of the chemical recycling process on composites.

Treated Sample	Starting Mass (g)	Recovered Fibers (g)	Recovered Polymer (g)	Recovery Yield (%)
Glass composite	10.97	7.05	3.43	95.5
Carbon composite	10.84	5.85	4.86	98.8

## Data Availability

The original contributions presented in the study are included in the article, further inquiries can be directed at the corresponding author.

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
