# Peer review of "Cleavable Bio-Based Epoxy Matrix for More Eco-Sustainable Thermoset Composite Components"

_polymers, 2024, doi:10.3390/polym17010088_

Round 1
Reviewer 1 Report
Comments and Suggestions for Authors
In the manuscript titled “Cleavable Bio-based Epoxy Matrix for More Eco-Sustainable Thermoset Composite Components”, the authors studied the material properties of a new type of bio-based thermoset composite. They also demonstrated that the composite is recyclable, allowing for the recovery of both the polymer matrix and reinforcing fibers. The fundamental results indicate that this system holds potential for practical application as a sustainable and recyclable material, although the specific area of application is not mentioned by the authors. The reviewer recommends the following revisions to the manuscript before it can be accepted for publication:
1. For the clarity of the authors, please mention the composition of the epoxy resin and hardener used in the present study.
2. Please mention the fiber volume/weight fraction used to fabricate the composite.
3. Pg 7, lines 264 – 266: “the first degradation step can be ascribed to …… during the first degradative step.” The conclusions drawn by the authors seem to lack clarity, making it difficult to fully understand. Please explain.
4. The authors report that the extent of increase in glass transition temperature is lower in carbon-reinforced composite compared to that in glass-reinforced composite. This observation contradicts the general trend observed in carbon/glass-reinforced materials where the carbon-reinforced composite usually has higher glass transition. What might the authors say to explain this?
5. Figure 11: I believe the TGA profile is of the recovered polymer from composite laminate post-cured at 100 oC. Is it correct? If so, please mention this.
6. The TGA profile of the recovered polymer shows only one major degradation (at ~315 – 340 oC) compared to the original polymer. Does this indicate some structural changes happening due to chemical recycling procedure? Is there a way to confirm that there are no significant structural changes happening?
7. In Sec. 3.6, the authors mention that the thermoset matrix is getting converted into a thermoplastic polymer on chemical recycling. For the clarity of the readers, please provide a schematic explanation, including chemical structures, to illustrate what the cleavage sites and the end polymer look like.
8. Was the re-fabrication of composite laminates using recycled polymer attempted?
9. How would the authors adjust the fiber weight fraction when fabricating a composite using the recycled polymer considering there is some contamination of the recovered thermoplastic with fibers to avoid significant material properties change?
10. Please check the text for incorrect verb tense, missing preposition (e.g., pg 1 line 27, pg 15 line 467, etc.), and typos.
Author Response
Reviewer 1
The authors would like to thank the reviewer for valuable comments and suggestions on this manuscript.
Below there are our replies to comments.
1. For the clarity of the authors, please mention the composition of the epoxy resin and hardener used in the
present study.
The comment has been implemented as requested in lines 149-164 with the addition of Figure 2a and 2b.
2. Please mention the fiber volume/weight fraction used to fabricate the composite.
The comment has been implemented as requested in lines 202-206.
3. Pg 7, lines 264 – 266: “the first degradation step can be ascribed to …… during the first degradative step.”
The conclusions drawn by the authors seem to lack clarity, making it difficult to fully understand. Please
explain.
The comment has been implemented as requested in lines 293-314.
4. The authors report that the extent of increase in glass transition temperature is lower in carbon-reinforced composite compared to that in glass-reinforced composite. This observation contradicts the general trend observed in carbon/glass-reinforced materials where the carbon-reinforced composite usually has higher glass transition. What might the authors say to explain this?
The comment has been implemented as requested in lines 604-611.
5. Figure 11: I believe the TGA profile is of the recovered polymer from composite laminate post-cured at 100
°C. Is it correct? If so, please mention this.
Yes, it is. The figure refers to the post-cured plate at 100 °C. I have added the reference to the text in lines 499500.
Please note: since figures were added in the manuscript, Figure 11, which you mentioned, has now
become Figure 12.
6. The TGA profile of the recovered polymer shows only one major degradation (at ~315 – 340 °C) compared
to the original polymer. Does this indicate some structural changes happening due to chemical recycling
procedure? Is there a way to confirm that there are no significant structural changes happening?
The explanation for this different degradation behavior has been included in lines 509-525. To support the
explanation, Figure 13 has also been added, showing the chemical structure of the recovered polymer.
7. In Sec. 3.6, the authors mention that the thermoset matrix is getting converted into a thermoplastic polymer
on chemical recycling. For the clarity of the readers, please provide a schematic explanation, including
chemical structures, to illustrate what the cleavage sites and the end polymer look like.
Figure 13 has been added, showing the chemical structure of the recovered polymer. Additionally, in
addressing the previous comment 6, multiple references have been included to further support the studied
material, which is still not well-known.
8. Was the re-fabrication of composite laminates using recycled polymer attempted?
The primary goal is to understand the processability of the recycled polymer, for example through FDM
printing. Producing laminated composites is a secondary step that requires a clear understanding of the
material obtained.
Cicala et al. [1] identified two applications for recovered thermoplastic: the production of tensile test
specimens through microinjection molding of a composite with kenaf fibers, and the fabrication of recycled
thermoplastic filaments for 3D printing using single-screw extrusion. Another work demonstrates the
printability of the material [2].
9. How would the authors adjust the fiber weight fraction when fabricating a composite using the recycled
polymer considering there is some contamination of the recovered thermoplastic with fibers to avoid
significant material properties change?
The issue of fiber contamination in the recovered thermoplastic primarily arises from the filtration system
used during the chemical recycling process. This system likely caused inefficiencies in fully separating the
dissolved polymer matrix from the residual fibers. We acknowledge that this aspect of the process requires
optimization. Notably, a study by Dattilo [3] in the literature, which conducted a similar recycling process and
characterized the recycled thermoplastic using TGA, did not report any residual fibers, emphasizing the
critical role of an optimized filtration system. Moving forward, we aim to implement a more advanced and
efficient filtration method to ensure complete fiber separation and eliminate contamination in the recycled
polymer.
10. Please check the text for incorrect verb tense, missing preposition (e.g., pg 1 line 27, pg 15 line 467, etc.),
and typos.
The comment has been implemented, and other detected errors have been corrected.
[1] Cicala, G., Pergolizzi, E., Piscopo, F., Carbone, D., & Recca, G. (2018). Hybrid composites manufactured by
resin infusion with a fully recyclable bioepoxy resin. Composites Part B: Engineering, 132, 69-76.
https://doi.org/10.1016/j.compositesb.2017.08.015
[2] Saitta, L., Montalbano, G., Corvaglia, I., Brovarone, C. V., & Cicala, G. (2023, October). Printability of a
Recycled Thermoplastic Obtained from a Chemical Recycling Process of a Fully‐Recyclable Epoxy Matrix: An
Upscaling Re‐Use Strategy. In Macromolecular Symposia (Vol. 411, No. 1, p. 2200188).
https://doi.org/10.1002/masy.202200188
[3] Dattilo, S., Cicala, G., Riccobene, P. M., Puglisi, C., & Saitta, L. (2022). Full recycling and re-use of bio-based
epoxy thermosets: chemical and thermomechanical characterization of the recycled matrices. Polymers, 14(22),
4828.
https://doi.org/10.3390/polym14224828

Reviewer 2 Report
Comments and Suggestions for Authors
In this manuscript, the effects of different post-curing conditions on thermal stability and glass transition temperature of resin were studied, and glass fiber and carbon fiber reinforced composites were prepared by vacuum-assisted resin transfer molding system and their recyclability was investigated. The topic is interesting and shows promising applications in the completely ecologically sustainable production and disposal of thermoset matrix composites. However, the manuscript is not well written and organized, it like a draft instead of a formal scientific paper. Therefore, it need to be revised carefully and thoroughly, specifically, following issues to be solved.
(1)The vacuum-assisted resin transfer molding system designed for laminate fabrication in Figure 2 is too vague. It is recommended to draw a clearer model diagram.
(2)The thermogravimetric image in Figure 4 requires data analysis using professional graphing software.
(3)Why the scale bar in the Figure 8(c) is different from the other figures in the Figure 8?
(4)The colors of the two legends in Figure 10 are too close to each other.
Comments on the Quality of English LanguageThe English could be improved to more clearly express the research
Author Response
Reviewer 2
The authors would like to thank the reviewer for valuable comments and suggestions on this manuscript.
Below there are our replies to comments.
1. The vacuum-assisted resin transfer molding system designed for laminate fabrication in Figure 2 is too
vague. It is recommended to draw a clearer model diagram.
The comment has been implemented by adding a schematic drawing of the vacuum infusion system, as shown
in Figure 3b. Additionally, references to the image have been included in the text for greater clarity.
2. The thermogravimetric image in Figure 4 requires data analysis using professional graphing software.
The image has been changed as suggested. Please note: since an additional image was inserted at the beginning
of the manuscript, Figure 4, which you referred to, has become Figure 5.
3. Why the scale bar in the Figure 8(c) is different from the other figures in the Figure 8?
Please note: Figure 8 has become Figure 9. The scale bar in figure is different because the image was taken at
a higher magnification compared to the other images in Figure 9. This was done intentionally to provide a
more detailed view of the specific features of the material that were not clearly visible at lower magnifications.
This clarification has been added in lines 413-414.
4. The colors of the two legends in Figure 10 are too close to each other.
The image has been modified, and the curves have been marked with more distinct colors for greater clarity,
as suggested. See Figure 11.

Round 2
Reviewer 2 Report
Comments and Suggestions for Authors
The issues are addressed in the revised manuscript